# CSVQ: Channel-wise Shared-Codebook Vector Quantization for Stable and Expressive Discrete Representations

## Abstract

Vector quantization (VQ) is a cornerstone of discrete representation learning, but existing methods often depend on very large codebooks that increase capacity while hurting stability and utilization. We present **Channel-wise Shared-codebook Vector Quantization (CSVQ)**, a simple tokenizer that quantizes each latent channel independently using one shared scalar codebook with per-channel normalization. CSVQ achieves the same representational capacity as vector-quantized latents while using a much smaller codebook, reduces gradient noise through channel-wise aggregation, and cuts codebook memory to scale linearly with the number of codewords rather than with both codewords and channels. Across multiple datasets and settings, CSVQ improves reconstruction quality and training stability, remains competitive under strict memory budgets, and scales favorably with model capacity. Finally, CSVQ shows improvements in reconstruction quality and downstream task performance compared to the state-of-the-art VQ methods. Ablation and multi-seed studies support the design choices. Code is available at https://anonymous.4open.science/r/csvq-CF06.

## 1 Introduction

Vector compression reduces memory, bandwidth, and latency by representing high-dimensional vectors with few bits. From a rate-distortion perspective, the aim is to maximize reconstruction quality under fixed resources. Vector Quantization (VQ) learns a discrete codebook that maps continuous latents to tokens, enabling memory-efficient storage, hardware-friendly nearest-neighbor lookup, and seamless use in token-based generative models. The VQ-VAE family (Van den Oord et al., 2017) shows that discrete latents can model complex data, yet jointly achieving high expressiveness and stable training remains challenging (Chang et al., 2022; Esser et al., 2021b; Razavi et al., 2019; Yu et al., 2021; Chen & Zhang, 2021; Kim & Park, 2022; Jackson & Miller, 2021; Martinez & Rodriguez, 2021).

Existing VQ methods rely on large codebooks to improve expressiveness, leading to training instability and codebook collapse where many entries remain unused Chen & Wang (2022); Kim & Park (2022); Lee & Choi (2023); Williams et al. (2020). (Williams et al., 2020) showed that large codebooks actually worsen convergence and stability, highlighting the need for more efficient designs Taylor & Anderson (2022); Johnson & Smith (2021).

Current research approaches can be categorized into two main strategies. First, holistic approaches quantize entire vectors as single units using large codebooks (Chang et al., 2022; Esser et al., 2021b; Razavi et al., 2019), but these methods exacerbate codebook collapse and instability issues (Van den Oord et al., 2017; Williams et al., 2020; Yu et al., 2021). Second, decomposition approaches employ multi-stage or dimensional partitioning Lee et al. (2022); Mentzer et al. (2023); O'Connor & Murphy (2021); Patel & Singh (2021), however, sequential dependencies and predefined structures limit optimization flexibility.

Despite progress, key gaps remain: 1) a common expressiveness criterion for fair comparison; 2) design principles that balance expressiveness and stability; 3) principled guidance for parameter selection; and 4) optimization strategies that retain decomposition benefits without strong structural constraints.

We propose **Channel-wise Shared-codebook Vector Quantization (CSVQ)**, which performs independent scalar quantization per channel using a single shared codebook preceded by channel-wise normalization. This removes structural coupling across dimensions, mitigates collapse, and achieves high expressiveness with small codebooks. Because all channels contribute to the same codebook, training signals are aggregated, improving stability.

Theoretically, we show CSVQ realizes $K^{(ch \times H \times W)}$ configurations where $K$ denotes the codebook size, $ch$ the number of latent channels, and $(H, W)$ the spatial resolution, while requiring only $O(K)$ codebook memory and reducing gradient variance by roughly $1/ch$. This yields clear trade-off rules under fixed budget, equal expressiveness, and channel-scaling regimes.

Empirically, CSVQ improves reconstruction on CIFAR-10 and ImageNet-64 under both fixed-budget and equal-expressiveness protocols, aligns with theory under channel scaling. Finally, CSVQ outperforms the state-of-the-art VQ methods for the final reconstruction benchmark and delivers stronger downstream performance (i.e., linear probing and transfer). CSVQ also exhibits better stability and utilization than baselines.

**Contributions.** (1) **Theory:** a unified expressiveness view for VQ methods, showing CSVQ achieves $K^{chHW}$ capacity with $O(K)$ memory and $\sim 1/ch$ gradient-variance reduction. (2) **Method:** a simple, stable channel-wise scalar quantizer with a shared codebook that alleviates collapse and scales gracefully with channels. (3) **Empirics & Practice:** comprehensive evaluation across controlled protocols and downstream tasks, with guidance for parameter selection under fixed budget and equal expressiveness.

## 2 RELATED WORK

### 2.1 VECTOR QUANTIZATION IN REPRESENTATION LEARNING

Vector Quantized Variational AutoEncoder (VQ-VAE) (Van den Oord et al., 2017) initiated discrete latent learning, showing complex distributions can be modeled with discrete codebooks rather than continuous latents. However, key challenges include codebook collapse (unused entries) and training instability. Williams et al. (2020) analyzed stabilization techniques like commitment loss weighting and EMA updates. SimpleVQ (SimVQ) (Zhu et al., 2024) prevents collapse by applying learnable linear transformations to frozen codebooks, providing a distinct approach to codebook utilization.

Additionally, methods like Vector Quantized Diffusion (VQ-Diffusion) model Gu et al. (2022) combine VQ with diffusion processes for enhanced generation quality, while Taming Transformers (Esser et al., 2021a) demonstrates VQ effectiveness in autoregressive generation. Each approach involves trade-offs between expressiveness, efficiency, and stability, with recent trends favoring simplified architectures that reduce training complexity.

### 2.2 MULTI-STAGE AND DECOMPOSITION QUANTIZATION

Multi-stage quantization approaches address single-stage VQ limitations. Lee et al. (2022) proposed Residual Quantized Variational Autoencoder (RQ-VAE) with a residual-based multi-layer structure, but introduced sequential dependencies causing bottlenecks. Product Quantization partitions vectors into sub-dimensions but has limited adaptability. Recent advances include Mentzer et al. (2023)'s Finite Scalar Quantization (FSQ) with fixed scalar units at different levels, and (Yu et al., 2023)'s Lookup Free Quantization (LFQ) employing lookup-free binary quantization with entropy regularization. These methods explore alternative quantization paradigms beyond traditional learnable codebooks.

### 2.3 THEORETICAL ANALYSIS OF DISCRETE REPRESENTATIONS

Discrete representation theory builds on Gray (1984)'s foundational vector quantization theory, establishing optimal continuous-to-discrete mappings. However, gaps exist between theoretical optimality and practical performance. High-dimensional quantization requires exponentially increasing codebook sizes Gersho & Gray (1992), creating expressiveness-complexity trade-offs. Pollard (1982) showed k-means convergence to local optima, while studies report unused entries causing memory waste Razavi et al. (2019).

Recent theoretical advances include Takida et al. (2022)'s analysis of codebook collapse mechanisms and Chou et al. (2002)'s entropy-constrained optimization frameworks. These works highlight the fundamental tension between codebook utilization and representation quality, motivating the design of more efficient quantization strategies.

### 2.4 POSITIONING OF CSVQ

Our CSVQ combines channel-wise independent scalar quantization with a single shared codebook to reduce structural coupling across dimensions. Unlike RQ-VAE (multi-stage complexity) and FSQ/LFQ (fixed schemes), it preserves spatial structure while achieving memory efficiency and expressiveness.

We extend classical VQ theory to exploit channel independence, analyzing expressiveness scaling and convergence. CSVQ offers a simpler, more scalable alternative to multi-stage architectures while remaining competitive with state-of-the-art methods.

## 3 PROPOSED METHOD

### 3.1 MOTIVATION

Vector quantization (VQ) maps continuous latent spaces to discrete symbols. For input $x \in \mathbb{R}^{H_0 \cdot W_0 \cdot 3}$ (i.e., $H_0 \times W_0$ for spatial resolution, 3 for RGB channels), encoder $E$ produces latent $z_e = E(x) \in \mathbb{R}^{H \cdot W \cdot ch}$. Each $ch$-dimensional vector $z_e^{(h,w)}$ maps to the nearest codebook entry $\mathcal{C} = \{c_k\}_{k=1}^K \subset \mathbb{R}^{ch}$:

$$z_q^{(h,w)} = \arg \min_{c_k \in \mathcal{C}} \|z_e^{(h,w)} - c_k\|_2. \tag{1}$$

Discrete representation expressiveness equals the total possible configurations: $N = K^{(H \cdot W)}$ when each spatial location independently selects from the codebook.

Existing VQ methods face three limitations: 1) High expressiveness requires $K = N^{1/(H \cdot W)}$, exponentially increasing memory $O(K \cdot ch)$, and search complexity $O(K \cdot ch)$. 2) Large codebooks suffer collapse with unused entries, causing memory waste and unstable training. 3) Memory scales with channels $ch$, limiting scalability.

CSVQ presents a new paradigm combining channel-wise independent quantization with shared scalar codebooks. It resolves structural constraints by processing channels independently while maintaining spatial structure. CSVQ achieves memory efficiency and expressiveness through shared codebooks, improving convergence via channel normalization and variance reduction through aggregation.

### 3.2 PROBLEM FORMULATION

We formulate discrete representation learning with encoder-decoder $(E_\phi, D_\theta)$ and quantizer $Q$ using finite alphabet $\mathcal{A}$ for $x \sim p_{\text{data}}$. The goal is to minimize reconstruction loss under rate/resource constraints.

**Constrained form.**

$$\min_{\phi, \theta, Q} \mathbb{E}_{x \sim p_{\text{data}}} \left[ \ell_{\text{rec}} \big( x, D_\theta(Q(E_\phi(x))) \big) \right] \quad \text{s.t. } \mathcal{R}(Q) \leq R_0, \ |\mathcal{A}| = K_0, \tag{2}$$

where $\mathcal{R}(Q)$ denotes rate (e.g., bits per site) or resource (memory/parameter) constraints.

**Lagrangian form.**

$$\min_{\phi, \theta, Q} \mathbb{E}_x \left[ \ell_{\text{rec}} \big( x, D_\theta(Q(E_\phi(x))) \big) + \lambda \cdot \mathcal{R}(Q) \right], \qquad \lambda \geq 0. \tag{3}$$

$\mathcal{R}(Q)$ can be defined as 1) symbol rate (e.g., bits per site $R$), 2) codebook size $K$, or 3) a proxy for memory/computation (e.g., codebook memory, nearest-neighbor (NN) search complexity), depending on the problem setting. The reconstruction loss uses $\ell_{\text{rec}} = \|x - \hat{x}\|_2^2$ or alternative losses that align with perceptual quality metrics.

Table 1: Notation and symbols used throughout the CSVQ formulation and theoretical analysis.

| Symbol | Definition |
|---|---|
| $x$ | Input image |
| $E_\phi, D_\theta$ | Encoder and decoder networks with parameters $\phi, \theta$ |
| $z_e \in \mathbb{R}^{H \cdot W \cdot d}$ | Latent representation (spatial resolution $H \cdot W$, channels $ch$) |
| $\tilde{z}_e$ | Channel-wise normalized latent representation |
| $z_q$ | Quantized latent representation |
| $\hat{x}$ | Reconstructed output $D_\theta(z_q)$ |
| $ch$ | Number of latent channels |
| $\mathcal{A}, K = |\mathcal{A}|$ | Finite alphabet (shared codebook) and its cardinality |
| $\mu_i, \sigma_i$ | Channel-wise normalization statistics (mean, std. deviation) |
| $Q$ | Quantization function mapping latents to discrete symbols |
| $R$ | Rate in bits per spatial site: $s \cdot \log_2 K$ for $s$ symbols |
| $\ell_{\text{rec}}$ | Reconstruction loss (typically $\|x - \hat{x}\|_2^2$) |
| $\beta$ | Commitment loss weighting parameter |
| $\gamma$ | EMA decay rate; $\varepsilon$: numerical stability constant |

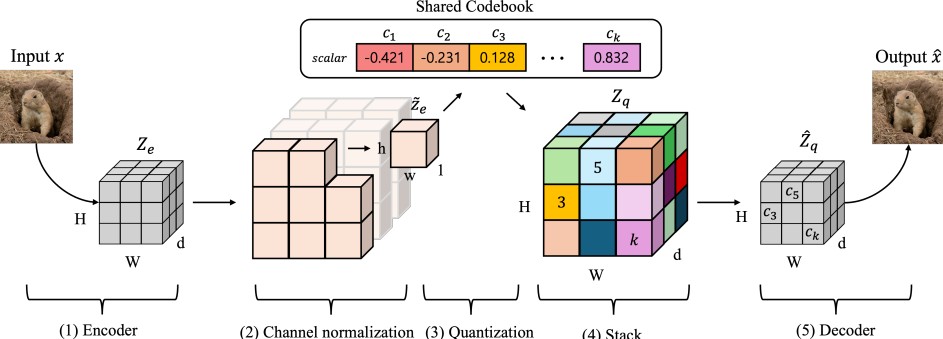

Figure 1: **CSVQ processing pipeline.** The method processes input images through five steps: (1) Encode input $x$ to latent representation $z_e \in \mathbb{R}^{H \times W \times ch}$; (2) Apply per-channel normalization; (3) Perform channel-wise scalar quantization using shared codebook $\mathcal{C}_{\text{shared}}$; (4) Stack quantized channels to form $\hat{Z}_q$; (5) Decode to reconstructed output $\hat{x}$.

**Symbols.** The encoder output is $z_e = E_\phi(x) \in \mathbb{R}^{H \cdot W \cdot ch}$, and $Q$ maps $z_e$ to symbols from the finite alphabet $\mathcal{A}$. The decoder reconstructs as $\hat{x} = D_\theta(Q(z_e))$. As an example of rate in this paper, we use $R = s \cdot \log_2 K$ when the number of symbols per site is $s$.

### 3.3 METHOD OVERVIEW

Figure 1 provides a high-level view of CSVQ. The next subsection formalizes the quantizer, the objective, and the training updates.

### 3.4 CHANNEL-WISE SHARED-CODEBOOK VECTOR QUANTIZATION (CSVQ)

**Channel-wise quantizer with normalization.** Given $z_e = E_\phi(x) \in \mathbb{R}^{H \times W \times ch}$, we compute per-channel statistics

$$\mu_i = \frac{1}{HW} \sum_{h,w} z_{e,h,w}^{(i)}, \qquad \sigma_i = \sqrt{\frac{1}{HW} \sum_{h,w} \left(z_{e,h,w}^{(i)} - \mu_i\right)^2 + \varepsilon},$$

and normalize $\tilde{z}_{e,h,w}^{(i)} = \left(z_{e,h,w}^{(i)} - \mu_i\right)/\sigma_i$. Each channel value is then quantized independently using a shared scalar codebook $\mathcal{C}_{\text{shared}} = \{c_k\}_{k=1}^K \subset \mathbb{R}$:

$$z_{q,h,w}^{(i)} = \arg \min_{c \in \mathcal{C}_{\text{shared}}} \left|\tilde{z}_{e,h,w}^{(i)} - c\right|, \quad i \in [ch], \ (h, w) \in [H] \times [W],$$

with a fixed tie-break to the smallest index when distances are equal. The quantized indices are decoded to codebook values $\hat{z}_{q,h,w}^{(i)} = c_{z_{q,h,w}^{(i)}}$ and we form $\hat{Z}_q = \text{stack}_i(\hat{z}_{q,:,:}^{(i)}) \in \mathbb{R}^{H \times W \times ch}$

**Shared codebook design.** Sharing $\mathcal{C}_{\text{shared}}$ across channels (1) reduces codebook memory from $O(K \cdot ch)$ to $O(K)$ (see Appendix B.5), (2) aggregates training signals across $ch$ channels, improving training stability (see Appendix C.2; proof in Appendix B.1), and (3) enforces consistent quantization granularity across channels. Per-site nearest-neighbor search is $O(ch \cdot K)$, but provides parallelization and caching due to dealing with effective simple scalars (see Appendix B.5).

**Objective and training updates.** We minimize

$$\min_{\phi, \theta, \mathcal{C}_{\text{shared}}} \mathbb{E}_{x \sim p_{\text{data}}} \left[ \underbrace{\|x - \hat{x}\|_2^2}_{\ell_{\text{rec}}} + \beta \underbrace{\sum_{i,h,w} \left( \tilde{z}_{e,h,w}^{(i)} - \text{sg}[\hat{z}_{q,h,w}^{(i)}] \right)^2}_{\ell_{\text{commit}}} \right], \quad \text{s.t. } |\mathcal{C}_{\text{shared}}| = K, \ \hat{x} = D_\theta(\hat{Z}_q),$$

where $\text{sg}[\cdot]$ denotes stop-gradient. We use the straight-through estimator (STE) for the encoder path and EMA for the codebook (convergence analysis in Appendix B.2):

$$N_k^{(t)} = \gamma N_k^{(t-1)} + (1 - \gamma) \sum_{i,h,w} \mathbf{1}\left[ z_{q,h,w}^{(i)} = k \right], \quad m_k^{(t)} = \gamma m_k^{(t-1)} + (1 - \gamma) \sum_{i,h,w: z_{q,h,w}^{(i)}=k} \tilde{z}_{e,h,w}^{(i)},$$

$$c_k \leftarrow \frac{m_k}{N_k + \varepsilon}.$$

## 3.5 THEORETICAL ANALYSIS

**Notation and assumptions.** A site refers to a single spatial location. Resolution is $H \cdot W$, number of channels is $ch$, codebook size is $K$, and RQ-VAE depth is $D$. Tie-breaking is handled with fixed tie-breaking rules, and channel/spatial independence is taken as an idealized assumption for configuration counting.

### 3.5.1 EXPRESSIVENESS AND RATE FAIRNESS

**Theorem 1** (Expressiveness). *Let $\mathcal{Q} : \mathbb{R}^{H \cdot W \cdot ch} \to \mathcal{S}$ be a quantization function mapping continuous latents to a discrete symbol space $\mathcal{S}$. For CSVQ with codebook size $K$, the cardinality of the symbol space is*

$$|\mathcal{S}_{CSVQ}| = K^{(ch \cdot H \cdot W)}.$$

*In comparison, VQ-VAE achieves $|\mathcal{S}_{VQ}| = K^{(H \cdot W)}$ and RQ-VAE with depth $D$ achieves $|\mathcal{S}_{RQ}| = K^{(D \cdot H \cdot W)}$.*

*Proof.* CSVQ performs independent scalar quantization at each $(i, h, w)$ position, where $i \in [ch]$, $h \in [H]$, $w \in [W]$. Each position has $K$ possible quantization outcomes, yielding $K^{(ch \cdot H \cdot W)}$ total configurations under independence. $\square$

**Corollary 1** (Equal-expressiveness mapping). *To achieve a target expressiveness $N = |\mathcal{S}|$, the required codebook sizes are:*

$$K_{VQ} = N^{1/(H \cdot W)}$$
$$K_{RQ} = N^{1/(D \cdot H \cdot W)}$$
$$K_{CSVQ} = N^{1/(ch \cdot H \cdot W)}$$

*For fixed $N$, CSVQ requires exponentially smaller codebook size when $ch > 1$.*

*Proof.* Direct consequence of Theorem 1 by solving $K^{\text{exponent}} = N$ for each method. $\square$

Table 2: Theoretical comparison of expressiveness, rate efficiency, and computational complexity across vector quantization methods.

| Method | Symbols/site | Configurations | Bits/site | CB Memory | Search Cost | Architecture |
|---|---|---|---|---|---|---|
| VQ-VAE | 1 vector ($ch$-dim) | $K^{H \cdot W}$ | $\log_2 K$ | $O(K \cdot ch)$ | $O(K \cdot ch)$ | Vector quantization |
| RQ-VAE | $D$ vectors | $K^{D \cdot H \cdot W}$ | $D \log_2 K$ | $O(K \cdot ch)$ | $O(D \cdot K \cdot ch)$ | Sequential residual |
| **CSVQ (Ours)** | $ch$ scalars | $\mathbf{K^{ch \cdot H \cdot W}}$ | $ch \log_2 K$ | $\mathbf{O(K)}$ | $O(ch \cdot K)$ | **Shared scalar CB** |

**Proposition 1** (Rate analysis). *For a target rate $R$ bits per spatial site, the codebook sizes required are:*

$$K_{VQ} = 2^R; K_{RQ} = 2^{R/D}; K_{CSVQ} = 2^{R/ch}.$$

*The rate achieved by each method is: $R_{VQ} = \log_2 K$, $R_{RQ} = D \log_2 K$, $R_{CSVQ} = ch \log_2 K$.*

*Proof.* Rate is defined as bits per spatial site. VQ-VAE uses one $\log_2 K$ bit symbol per site. RQ-VAE uses $D$ symbols requiring $D \log_2 K$ bits. CSVQ uses $ch$ scalar symbols requiring $ch \log_2 K$ bits. □

Table 2 comprehensively summarizes these theoretical results. Our CSVQ requires a smaller codebook size $K$ to achieve the same expressiveness and is more efficient than existing methods with memory complexity $O(K)$.

### 3.5.2 COMPUTATIONAL AND MEMORY COMPLEXITY

**Proposition 2** (Computational complexity). *The per-site computational and memory complexities are:*

| Method | Search Complexity | Memory Complexity |
|---|---|---|
| *VQ-VAE* | $O(K \cdot ch)$ | $O(K \cdot ch)$ |
| *RQ-VAE* | $O(D \cdot K \cdot ch)$ | $O(K \cdot ch)$ |
| *CSVQ* | $O(K \cdot ch)$ | $O(K)$ |

*CSVQ achieves $ch$-fold memory reduction while maintaining comparable search complexity with superior parallelization properties.*

*Proof.* VQ-VAE stores $K$ vectors of dimension $ch$, requiring $O(K \cdot ch)$ memory and $O(K \cdot ch)$ distance computations per site. RQ-VAE performs $D$ sequential searches with $O(K \cdot ch)$ memory total. CSVQ stores $K$ scalars requiring $O(K)$ memory, performing $ch$ independent scalar searches of $O(K)$ each, totaling $O(K \cdot ch)$ with superior cache locality. Full proof is provided in Appendix B.5. □

Our theoretical analysis relies on channel independence, which we justify through the architectural design that processes each channel independently rather than jointly quantizing the entire channel vector, enabling independent scalar quantization at each channel.

**Summary:** 1) Configuration count comparison is based on independent selection and tie-breaking idealization assumptions. 2) Rate analysis provides a theoretical foundation for comparing methods under equivalent bit-rate constraints. 3) Actual performance is influenced by codebook utilization, distribution imbalance, dead entries, etc., so we report Equal-Expressiveness experiments to validate theoretical predictions.

## 4 EXPERIMENTS

### 4.1 EXPERIMENTAL SETUP

**Datasets:** Primary reconstruction and analysis are performed on CIFAR-10 ($32 \times 32$, 50K images) and ImageNet-64 ($64 \times 64$, 1.28M images). For downstream evaluation, we use STL-10 ($32 \times 32$, 5K) and ImageNet Subset-10 ($256 \times 256$, 128K) to verify domain shift and generalization.

**Evaluation Metrics:** Reconstruction quality is evaluated using Peak Signal-to-Noise Ratio (PSNR) and Structural Similarity Index (SSIM), with bits-per-pixel (bpp) reported as a compression metric when needed.

**Baselines:** To isolate the effect of our design choices, we primarily compare against the canonical VQ family with learnable, discrete codebooks—VQ-VAE Van den Oord et al. (2017) and RQ-VAE Lee et al. (2022). We additionally include recent alternatives with distinct quantization paradigms (FSQ Mentzer et al. (2023), SIM-VQ Zhu et al. (2024), LFQ Yu et al. (2023)) to support a final reconstruction benchmark and to assess applicability to downstream tasks.

## 4.2 EVALUATION PROTOCOLS

We compare methods under complementary protocols that isolate different trade-offs. Let $d$ be the number of latent channels, $D$ the RVQ depth, $H \times W$ the latent spatial size, and $K$ the codebook size.

**P1: Fixed Codebook Memory.** Fix a codebook memory budget $M_{cb}$ (number of scalar parameters). We set CSVQ: $K = M_{cb}$, VQ-VAE: $K = \frac{M_{cb}}{ch}$, RQ-VAE: $K = \frac{M_{cb}}{ch}$. This measures quality per unit memory using each method's natural codebook structure.

**P2: Equal Expressiveness.** Fix a target configuration count $N$ and choose $(K, ch, D)$ so that $K^{ch \cdot H \cdot W}$ (CSVQ), $K^{H \cdot W}$ (VQ-VAE), and $K^{D \cdot H \cdot W}$ (RQ-VAE) all equal $N$. To exclude dimensional effects, all methods use scalar codebooks in this protocol.

**P3: Channel Scaling.** Fix $K$ (and $D$ for RQ-VAE) and increase channels $ch \to ch'$. Expressiveness scales as $K^{ch \cdot H \cdot W} \to K^{ch' \cdot H \cdot W}$ for CSVQ, while vector VQ remains $K^{H \cdot W}$, revealing channel-axis scalability.

Finally, we define **P4-Bench** as a practical benchmark setting that matches codebook size $K$ and channels $ch$ across methods to support final reconstruction benchmarking and downstream applicability assessment.

Full definitions, assumptions, and rate mappings are provided in Appendix A.3.

## 4.3 EXPERIMENTAL METHODOLOGY

### 4.3.1 FIXED CODEBOOK BUDGET COMPARISON (P1)

Table 3: Reconstruction performance under fixed parameter budget (P1).

| Dataset | Method | $M_{cb}$ | CB | Ch | D | Expressiveness | PSNR↑ | SSIM↑ | bpp↓ |
|---|---|---|---|---|---|---|---|---|---|
| CIFAR-10 | VQ-VAE | $16 \cdot 4$ | 16 | 4 | - | $16^{8 \times 8}$ | 10.99 | 0.6794 | 0.0625 |
| | RQ-VAE | $16 \cdot 4$ | 16 | 4 | 4 | $16^{4 \times 8 \times 8}$ | 19.37 | 0.9639 | 0.25 |
| | RQ-VAE | $16 \cdot 4$ | 16 | 4 | 8 | $16^{8 \times 8 \times 8}$ | 19.08 | 0.9619 | 0.5 |
| | CSVQ (Ours) | $64 \cdot 1$ | 64 | 4 | - | $64^{4 \times 8 \times 8}$ | **20.50** | **0.9735** | 0.25 |
| ImageNet-64 | VQ-VAE | $16 \cdot 4$ | 16 | 4 | - | $16^{8 \times 8}$ | 8.74 | 0.4122 | 0.0625 |
| | RQ-VAE | $16 \cdot 4$ | 16 | 4 | 4 | $16^{4 \times 8 \times 8}$ | 17.14 | 0.9137 | 0.25 |
| | RQ-VAE | $16 \cdot 4$ | 16 | 4 | 8 | $16^{8 \times 8 \times 8}$ | 16.88 | 0.9092 | 0.5 |
| | CSVQ (Ours) | $64 \cdot 1$ | 64 | 4 | - | $64^{4 \times 8 \times 8}$ | **17.84** | **0.9265** | 0.25 |

Under protocol P1 with a fixed 64-parameter budget, we match the codebook parameter budget $M_{cb}$ as (codebook size × codebook dimension): VQ/RQ use $16 \times 4$ (vector codebooks) and CSVQ uses $64 \times 1$ (scalar codebook), yielding the same $M_{cb}=64$ for a fair comparison. To align model structures, CSVQ also uses 4 channels, and RQ-VAE is set to depth 4 accordingly; since RQ-VAE can scale computation via depth, we additionally report depth 8 under the same memory budget. On CIFAR-10, CSVQ achieves 20.50 PSNR versus VQ-VAE's 10.99 and RQ-VAE's 19.37. On ImageNet-64, CSVQ reaches 17.84 versus VQ-VAE's 8.74 and RQ-VAE's 17.14, demonstrating superior resource efficiency.

For compression efficiency, VQ-VAE shows the lowest bpp (0.0625) but poor performance. RQ-VAE and CSVQ both use 0.25 bpp, but CSVQ achieves superior quality, confirming better expressiveness utilization.

### 4.3.2 Equal Expressiveness Settings (P2)

Table 4: Equal expressiveness comparison (P2).

| Method | CB | Ch | D | Expressiveness | CIFAR-10 | | ImageNet-64 | |
| --- | --- | --- | --- | --- | --- | --- | --- | --- |
| | | | | | PSNR↑ | SSIM↑ | PSNR↑ | SSIM↑ |
| VQ-VAE | 512 | 1 | - | $512^{8\times8}$ | 15.90 | 0.9170 | 14.77 | 0.8517 |
| RQ-VAE | 8 | 1 | 3 | $512^{8\times8}$ | 13.58 | 0.8822 | 13.24 | 0.8030 |
| CSVQ (Ours) | **8** | **3** | - | $512^{8\times8}$ | **17.99** | **0.9504** | **16.40** | **0.8977** |
| VQ-VAE | 729 | 1 | - | $729^{8\times8}$ | 15.62 | 0.9104 | 14.67 | 0.8489 |
| RQ-VAE | 9 | 1 | 3 | $729^{8\times8}$ | 13.80 | 0.8816 | 13.21 | 0.8058 |
| CSVQ (Ours) | **9** | **3** | - | $729^{8\times8}$ | **18.18** | **0.9527** | **16.36** | **0.8968** |
| VQ-VAE | 4096 | 1 | - | $4096^{8\times8}$ | 15.98 | 0.9209 | 14.81 | 0.8523 |
| RQ-VAE | 8 | 1 | 4 | $4096^{8\times8}$ | 13.49 | 0.8776 | 13.01 | 0.7982 |
| CSVQ (Ours) | **8** | **4** | - | $4096^{8\times8}$ | **19.08** | **0.9622** | **16.98** | **0.9112** |
| RQ-VAE | 512 | 1 | 3 | $512^{3\times8\times8}$ | 15.45 | 0.9070 | 14.65 | 0.8422 |
| CSVQ (Ours) | **512** | **3** | - | $512^{3\times8\times8}$ | **19.22** | **0.9636** | **17.02** | **0.9109** |

Protocol P2 tests equal expressiveness $512^{8\times8}$. Table 4 shows CSVQ achieves superior performance with smaller codebooks. CSVQ (codebook 8) outperforms VQ-VAE (512) and RQ-VAE (8, depth 3). Moreover, scaling either the codebook size or the channel(depth) consistently favors CSVQ under matched expressiveness; among the two knobs, increasing channels yields larger expressiveness growth than merely enlarging the codebook and translates into stronger reconstruction improvements. We further verify large-codebook regimes by increasing codebook sizes under matched expressiveness, where CSVQ (512) still outperforms competing methods.

### 4.3.3 Channel Scaling Analysis (P3)

Protocol P3 tests channel scalability with fixed codebook sizes. We include a non-quantized autoencoder (AE) as a continuous upper bound to contextualize comparisons among quantization methods. Table 5 confirms CSVQ's continuous improvement with increasing channels. At 64 channels on CIFAR-10, CSVQ reaches 29.48 PSNR versus VQ-VAE's 9.40 and RQ-VAE's 19.54. CSVQ scales proportionally with channels while baseline methods plateau, demonstrating superior scalability relative to vector-quantized baselines.

### 4.3.4 Reconstruction Benchmark (P4)

Building on the controlled comparisons in Protocols P1–P3, we define a final reconstruction benchmark under Protocol P4 (codebook 256 and channels 32). Although all methods share the same nominal parameters for comparability, their quantization mechanisms differ: FSQ ($\{8, 8, 4\}$ scalar levels), LFQ ($\{-1, +1\}^8$ binary encoding), and SIM-VQ (learned transforms over 256 frozen Gaussian vectors). We evaluate on CIFAR-10 and ImageNet subset-10 to assess reconstruction quality on both low and high-resolution datasets, with ImageNet subset-10 providing continuity with downstream task evaluation.

Table 6 summarizes reconstruction results (PSNR/SSIM). CSVQ achieves the best reconstruction under P4, with large margins on CIFAR-10 (33.05 PSNR, 0.9989 SSIM) and consistent gains on ImageNet subset-10 (14.05 PSNR, 0.8533 SSIM), outperforming all other methods.

### 4.3.5 Downstream Evaluation (P4)

We evaluate representation quality through linear probing and transfer learning using Protocol P4 (codebook size 256, channels 32). Encoders are trained on CIFAR-10 and ImageNet subset-10 and kept frozen during downstream tasks.

**Linear Probing:** Train a linear classifier on flattened latents with the encoder frozen; use the encoder's training dataset.

Table 5: Channel scalability analysis (P3).

| Dataset | Method | CB | D | PSNR | | | SSIM | | |
|---|---|---|---|---|---|---|---|---|---|
| | | | | ch=8 | ch=64 | diff | ch=8 | ch=64 | diff |
| CIFAR-10 | AE (Upper bound) | - | - | 25.94 | 45.41 | +19.47 | 0.9935 | 1.0000 | +0.0065 |
| | VQ-VAE | 16 | - | 12.35 | 9.13 | -3.22 | 0.8000 | 0.4572 | -0.3428 |
| | RQ-VAE | 16 | 8 | 19.81 | 20.10 | +0.29 | 0.9682 | 0.9701 | +0.0019 |
| | CSVQ (Ours) | 16 | - | **23.25** | **31.52** | **+8.27** | **0.9870** | **0.9981** | **+0.0111** |
| ImageNet-64 | AE (Upper bound) | - | - | 20.60 | 34.09 | +13.49 | 0.9622 | 0.9988 | +0.0366 |
| | VQ-VAE | 32 | - | 11.38 | 9.40 | -1.98 | 0.6668 | 0.4859 | -0.1809 |
| | RQ-VAE | 32 | 8 | 18.76 | 19.54 | +0.78 | 0.9406 | 0.9509 | +0.0103 |
| | CSVQ (Ours) | 32 | - | **19.87** | **29.48** | **+9.61** | **0.9551** | **0.9958** | **+0.0407** |

Table 6: Reconstruction benchmark and downstream task evaluation (P4).

| Method | Reconstruction | | | Linear Probing | | Transfer Learning | | |
|---|---|---|---|---|---|---|---|---|
| | Enc. Data | PSNR↑ | SSIM↑ | Top-1↑ | Top-3↑ | Task Data | Top-1↑ | Top-3↑ |
| VQ-VAE | | 15.05 | 0.8906 | 36.79 | 68.84 | | 33.67 | 65.74 |
| RQ-VAE | | 24.22 | 0.9892 | 46.42 | 77.18 | | 39.38 | 70.51 |
| FSQ | CIFAR-10 | 18.35 | 0.9538 | 40.22 | 70.68 | STL-10 | 36.33 | 67.79 |
| SIM-VQ | | 10.26 | 0.5915 | 26.86 | 58.79 | | 24.45 | 53.16 |
| LFQ | | 6.25 | 0.0760 | 23.92 | 51.74 | | 22.11 | 48.88 |
| CSVQ (Ours) | | **33.05** | **0.9989** | **48.63** | **79.43** | | **45.05** | **75.33** |
| VQ-VAE | | 12.70 | 0.7844 | 39.52 | 68.15 | | 34.46 | 60.16 |
| RQ-VAE | | 13.75 | 0.8337 | 42.34 | 70.16 | | 34.26 | 60.96 |
| FSQ | ImageNet | 12.73 | 0.7890 | 38.71 | 65.93 | ImageNet | 32.07 | 59.76 |
| SIM-VQ | subset-10 | 10.24 | 0.6587 | 29.84 | 63.10 | subset-10 | 30.68 | 60.16 |
| LFQ | | 5.50 | 0.0738 | 15.12 | 39.52 | | 20.12 | 47.21 |
| CSVQ (Ours) | | **14.05** | **0.8533** | **45.97** | **73.79** | | **38.05** | **67.73** |

**Transfer Learning:** Freeze the pre-trained encoder and train a linear head on STL-10 or ImageNet subset-10 (disjoint classes).

Table 6 shows that CSVQ achieves superior downstream performance across both linear probing and transfer learning tasks, with consistent improvements over all baselines, demonstrating that its discrete representations preserve information beneficial for semantic tasks.

# 5 CONCLUSION

We introduced **CSVQ**, a channel-wise scalar quantizer with a *shared* codebook that reduces memory and stabilizes training while preserving (and often improving) reconstruction quality. Our analysis shows CSVQ attains expressiveness $K^{(ch \cdot H \cdot W)}$ with codebook memory $O(K)$ and lowers gradient variance by $\Theta(1/ch)$, providing clear guidance for equal-rate and equal-expressiveness settings. Across CIFAR-10 and ImageNet-64, CSVQ outperforms VQ-VAE/RQ-VAE under fixed budget and matched expressiveness, scales favorably with channels, and yields stronger downstream linear-probe/transfer results; codebook perplexity and dead-entry metrics corroborate better utilization.

These findings position CSVQ as a simple, scalable, and resource-efficient alternative for discrete representation learning. Future work includes integrating CSVQ into autoregressive generators, adapting codebooks dynamically under rate–distortion constraints, and extending analysis to temporal and multimodal data.

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
