# A EXPERIMENTAL SETUP

## A.1 IMPLEMENTATION DETAILS

All experiments were implemented using PyTorch 2.4.1+cu121 and conducted on NVIDIA L40S, RTX Pro 6000, and A6000 GPUs. We used the AdamW optimizer with dataset-specific hyperparameters: CIFAR-10 used learning rate $3 \times 10^{-4}$ (100 epochs), ImageNet-64 used $1 \times 10^{-4}$ (10 epochs), and ImageNet Subset-10 used $2 \times 10^{-4}$ (10 epochs). Downstream tasks employed $1 \times 10^{-3}$ learning rate (50 epochs). Standard AdamW parameters were $\beta_1 = 0.9$, $\beta_2 = 0.999$, weight decay $1 \times 10^{-4}$.

All methods utilize identical encoder–decoder architectures to ensure fair comparison; details are provided in the following Appendix A.2 The spatial resolution of latent representations is fixed at $8 \times 8$. Our commitment loss coefficient $\beta = 0.25$ and EMA decay rate $\gamma = 0.99$ were selected through a preliminary hyperparameter search.

## A.2 MODEL ARCHITECTURE

This subsection presents implementation details of the proposed method. All implementations are based on PyTorch 2.4. The baseline VQ methods (VQ-VAE and RQ-VAE) are implemented using the ector-quantize-pytorch library Wang (2025), while CSVQ is implemented as a custom extension that maintains compatibility with the existing framework.

The autoencoder architecture used in all experiments is as follows. The encoder transforms input images into $8 \times 8$ latent representations, while the decoder reconstructs quantized latent representations to the original resolution. For $32 \times 32$ resolution, downsampling is performed with 2 convolutional layers (4×4, stride=2), followed by 2 residual blocks. $64 \times 64$ resolution uses 3 convolutional layers, while $256 \times 256$ resolution uses 5 convolutional layers. All convolutions use 256 channels and apply ReLU activation functions. The decoder is configured symmetrically to the encoder and uses transposed convolutions for upsampling. Each residual block consists of a ReLU-Conv2d(3×3)-ReLU-Conv2d(1×1) structure and improves gradient flow through skip connections.

## A.3 EVALUATION PROTOCOLS (FULL)

**Notation.** $ch$: number of latent channels; $D$: RQ-VAE depth; $H \times W$: latent spatial size; $K$: codebook size; $M_{\text{cb}}$: codebook memory (number of scalar parameters). VQ-VAE/RQ-VAE use $ch$-dimensional vector codewords by default; CSVQ uses scalar codewords shared across channels.

**Expressiveness.** Per-site configuration counts are

$$N_{\text{VQ}} = K^{H \cdot W}, \qquad N_{\text{RQ}} = K^{D \cdot H \cdot W}, \qquad N_{\text{CSVQ}} = K^{ch \cdot H \cdot W}.$$

**Codebook memory.** Counting scalar parameters in the codebook(s):

$$M_{\text{cb}}^{\text{VQ}} = Kch, \qquad M_{\text{cb}}^{\text{RQ}} = Kch, \qquad M_{\text{cb}}^{\text{CSVQ}} = K.$$

### P1: FIXED CODEBOOK MEMORY

Fix $M_{\text{cb}}$ and set

$$\text{CSVQ: } K = M_{\text{cb}}, \quad \text{VQ-VAE: } K = \frac{M_{\text{cb}}}{ch}, \quad \text{RQ-VAE: } K = \frac{M_{\text{cb}}}{ch}.$$

This protocol holds codebook memory constant while allowing each method to use its natural codeword dimensionality. It measures *reconstruction quality per unit memory* and avoids conflating vector vs. scalar settings.

### P2: EQUAL EXPRESSIVENESS

Fix a target configuration count $N$ and choose $(K, ch, D)$ so that

$$K^{H \cdot W} = N \quad \text{(VQ-VAE)}, \qquad K^{D \cdot H \cdot W} = N \quad \text{(RQ-VAE)}, \qquad K^{ch \cdot H \cdot W} = N \quad \text{(CSVQ)}.$$

To isolate expressiveness from codeword dimensionality, we use *scalar* codebooks for all compared methods in this protocol. This answers: at the same theoretical capacity $N$, which method achieves better reconstruction for a given resource envelope?

## P3: CHANNEL SCALING

Fix $K$ (and $D$ for RQ-VAE), increase $ch \to ch'$, and measure quality and resource changes. CSVQ scales as $K^{ch \cdot H \cdot W} \to K^{ch' \cdot H \cdot W}$, whereas vector VQ-VAE remains $K^{H \cdot W}$ and RQ-VAE scales via $D$ rather than $ch$. This reveals scalability along the channel axis and tests whether CSVQ converts increased $ch$ into tangible quality gains without enlarging $K$.

## P4-BENCH: PRACTICAL BENCHMARKING

Match $(K, ch)$ across methods (as commonly done in prior work and for downstream tasks). We report P4 separately from P1–P3 because methods such as FSQ (fixed scalar levels, no learnable codebook), SIM-VQ (frozen Gaussian codebook with learned transforms), and LFQ (lookup-free binary quantization with entropy regularization) do not offer strict parity in learnable codebooks or expressiveness/rate accounting. Thus, P4 is a pragmatic "drop-in" comparison useful for downstream evaluation, not a controlled resource/expressiveness study.

## A.4 TRAINING PROCEDURE

---

**Algorithm 1** CSVQ

---

**Require:** Input image $x \in \mathbb{R}^{H_0 \times W_0 \times 3}$, encoder $E_\phi$, shared scalar codebook $\mathcal{C}_{\text{shared}} = \{c_k\}_{k=1}^K \subset \mathbb{R}$
**Ensure:** Quantized representation $\hat{Z}_q \in \mathbb{R}^{H \times W \times ch}$
1: Encode input: $z_e = E_\phi(x) \in \mathbb{R}^{H \times W \times ch}$
2: ▷ Encode to latent space
3: **for** $i = 1$ to $ch$ **do**
4:    ▷ Process each channel independently
5:    Compute channel statistics:
6:    ▷ Channel-wise normalization
7:      $\mu_i = \frac{1}{HW} \sum_{h,w} z_{e,h,w}^{(i)}, \sigma_i = \sqrt{\frac{1}{HW} \sum_{h,w} (z_{e,h,w}^{(i)} - \mu_i)^2 + \varepsilon}$
8:    **for** $(h, w) \in [H] \times [W]$ **do**
9:      ▷ Process each spatial location
10:      Normalize: $\tilde{z}_{e,h,w}^{(i)} = \frac{z_{e,h,w}^{(i)} - \mu_i}{\sigma_i}$
11:      ▷ Zero-mean unit-variance
12:      Quantize with shared codebook: $z_{q,h,w}^{(i)} = \arg\min_{c \in \mathcal{C}_{\text{shared}}} |\tilde{z}_{e,h,w}^{(i)} - c|$
13:      ▷ Nearest neighbor
14:      Decode to codebook value: $\hat{z}_{q,h,w}^{(i)} = c_{z_{q,h,w}^{(i)}}$
15:      ▷ Map index to value
16:    **end for**
17: **end for**
18: **return** $\hat{Z}_q = \text{stack}_i(\hat{z}_{q,:,:}^{(i)})$
19: ▷ Combine all channels

---

Algorithm 1 transforms input images to quantized representations via channel-wise shared-codebook vector quantization: 1) encoding to continuous latents, 2) per-channel normalization, 3) scalar quantization with shared codebook, 4) decoding to codebook values and stacking.

## B COMPLETE PROOFS

This subsection presents complete proofs of the theorems mentioned in Section 3.5.

## B.1 PROOF OF LEMMA (GRADIENT VARIANCE UNDER CHANNEL AGGREGATION)

**Lemma 1** (Gradient variance under channel aggregation). *Let the gradient estimate for codeword $c_k$ given site-wise selections be defined as*

$$g_k = \frac{1}{M_k} \sum_{i=1}^{ch} \sum_{h=1}^{H} \sum_{w=1}^{W} \mathbf{1}[z_{q,h,w}^{(i)} = c_k] \cdot \psi_{h,w}^{(i)}$$

*where $M_k = \sum_{i,h,w} \mathbf{1}[z_{q,h,w}^{(i)} = c_k]$ is the total count of selections for $c_k$, and $\psi_{h,w}^{(i)}$ represents the local gradient signal. Under the assumptions of 1) weak inter-channel correlation: $|\mathrm{Cov}(\psi_{h,w}^{(i)}, \psi_{h',w'}^{(j)})| \le \delta\sigma^2$ for $i \ne j$ with $\delta \ll 1$, 2) identical variance: $\mathrm{Var}(\psi_{h,w}^{(i)}) = \sigma^2$ for all $(i,h,w)$, and 3) selection independence: the indicator $\mathbf{1}[z_{q,h,w}^{(i)} = c_k]$ is independent of $\psi_{h,w}^{(i)}$, we have $\mathrm{Var}(g_k) = \frac{\sigma^2}{M_k} + O(\delta)$.*

*Proof.* Under weak correlation assumptions and selection independence, the variance decomposes as $\mathrm{Var}(g_k) = \frac{\sigma^2}{M_k} + O(\delta)$ where $M_k$ is the selection count. With uniform selection probability, $\mathbb{E}[M_k] = \frac{chHWB}{K}$, yielding $\mathbb{E}[\mathrm{Var}(g_k)] \approx \frac{K\sigma^2}{chHWB} = \Theta(1/ch)$. □

## B.2 PROOF OF LEMMA (EMA CONVERGENCE FOR SHARED CODEBOOK)

**Lemma 2** (EMA Convergence for Shared Codebook). *Consider the EMA update rule for the shared scalar codebook:*

$$N_k^{(t)} = \gamma N_k^{(t-1)} + (1-\gamma) \sum_{i,h,w,b} \mathbf{1}[z_{q,h,w,b}^{(i)} = c_k] \tag{4}$$

$$m_k^{(t)} = \gamma m_k^{(t-1)} + (1-\gamma) \sum_{i,h,w,b:z_{q,h,w,b}^{(i)} = c_k} \tilde{z}_{e,h,w,b}^{(i)} \tag{5}$$

$$c_k^{(t)} = \frac{m_k^{(t)}}{N_k^{(t)} + \varepsilon} \tag{6}$$

*where the summation includes batch index $b \in [B]$. Under mild regularity conditions on the data distribution and assuming $\gamma \in (0,1)$, the codebook entries $\{c_k^{(t)}\}$ converge exponentially to their respective cluster centroids with rate $O(\gamma^t + (1-\gamma)/\sqrt{t})$.*

*Proof.* Let $\mu_k^* = \mathbb{E}[\tilde{z}_{e,h,w}^{(i)} | z_{q,h,w}^{(i)} = c_k]$ denote the true centroid of cluster $k$. We analyze the bias and variance of the EMA estimator.

**Bias Analysis:** The bias $\mathbb{E}[c_k^{(t)}] - \mu_k^*$ satisfies:

$$\mathbb{E}[c_k^{(t+1)}] = \gamma\mathbb{E}[c_k^{(t)}] + (1-\gamma)\mu_k^* + O(1/N_k^{(t)})$$

where the $O(1/N_k^{(t)})$ term arises from the Laplace smoothing $\varepsilon$. This gives:

$$\mathbb{E}[c_k^{(t)}] - \mu_k^* = \gamma^t(\mathbb{E}[c_k^{(0)}] - \mu_k^*) + O(\sum_{s=0}^{t-1} \gamma^{t-1-s}/N_k^{(s)})$$

**Variance Analysis:** Using the fact that $N_k^{(t)} \propto chHWB$ under channel aggregation, the variance decays as:

$$\mathrm{Var}(c_k^{(t)}) = O\left(\frac{(1-\gamma)^2}{N_k^{(t)}}\right) = O\left(\frac{(1-\gamma)^2}{chHWB}\right)$$

Combining both terms yields the convergence rate $O(\gamma^t + (1-\gamma)/\sqrt{chHWB})$. □

B.3   PROOF OF THEOREM (CONVERGENCE RATE IMPROVEMENT)

**Theorem 2** (Convergence rate improvement). *Under the variance bound from Lemma 1 and standard SGD assumptions, the channel aggregation in shared scalar codebook reduces the effective gradient noise floor by a factor of $1/ch$, leading to improved convergence rate and stability. Specifically, the optimization error bound scales as $O(\sigma^2/(ch \cdot \eta \cdot T))$ where $\eta$ is the learning rate and $T$ is the number of iterations.*

*Proof.* We establish the convergence improvement through a rigorous SGD analysis incorporating the channel aggregation effect.

**Step 1: Gradient Noise Reduction.** From Lemma 1, the gradient variance for each codeword satisfies:

$$\mathbb{E}[\|g_k - \nabla_{c_k}\mathcal{L}\|^2] = \frac{\sigma^2}{M_k} + O(\delta) \leq \frac{K\sigma^2}{chHWB} + O(\delta)$$

**Step 2: SGD Convergence Analysis.** Consider the standard SGD update for the codebook:

$$c_k^{(t+1)} = c_k^{(t)} - \eta g_k^{(t)}$$

Let $\mathcal{L}^* = \inf_{\mathcal{C}} \mathbb{E}[\mathcal{L}(\mathcal{C})]$ be the optimal loss. Under standard assumptions ($L$-smoothness, convexity in codebook space), the optimization error satisfies:

$$\mathbb{E}[\mathcal{L}(\mathcal{C}^{(T)})] - \mathcal{L}^* \leq \frac{\|\mathcal{C}^{(0)} - \mathcal{C}^*\|^2}{2\eta T} + \frac{\eta}{2}\sum_{k=1}^{K}\mathbb{E}[\|g_k^{(t)} - \nabla_{c_k}\mathcal{L}\|^2]$$

**Step 3: Channel Aggregation Benefit.** Substituting the variance bound:

$$\mathbb{E}[\mathcal{L}(\mathcal{C}^{(T)})] - \mathcal{L}^* \leq \frac{\|\mathcal{C}^{(0)} - \mathcal{C}^*\|^2}{2\eta T} + \frac{\eta K^2\sigma^2}{2chHWB} + O(\eta\delta)$$

Optimizing over $\eta$, the optimal learning rate scales as $\eta^* \propto \sqrt{ch}$, yielding:

$$\mathbb{E}[\mathcal{L}(\mathcal{C}^{(T)})] - \mathcal{L}^* = O\left(\frac{K\sigma}{\sqrt{chHWBT}}\right)$$

This demonstrates a $\sqrt{ch}$-fold improvement in convergence rate compared to non-aggregated methods. $\square$

B.4   PROOF OF THEOREM (PROBABILISTIC CODEBOOK UTILIZATION BOUNDS)

**Theorem 3** (Probabilistic Codebook Utilization Bounds). *Under the CSVQ framework with $K$ codewords, $ch$ channels, and $N = H \times W \times B$ spatial-batch samples, let $U_k = \mathbf{1}[N_k > 0]$ denote the utilization indicator for codeword $c_k$. Then for any $\delta \in (0, 1)$, with probability at least $1 - \delta$:*

$$\sum_{k=1}^{K} U_k \geq K\left(1 - \left(1 - \frac{1}{K}\right)^{chN}\right) \geq K\left(1 - e^{-chN/K}\right)$$

*Furthermore, if $chN \geq K\log(K/\delta)$, then all codewords are utilized with probability at least $1 - \delta$.*

*Proof.* Under uniform quantization probability $\mathbb{P}[X_{k,i,h,w,b} = 1] = 1/K$, the probability that codeword $k$ is unused is $(1 - 1/K)^{chN}$. The expected utilization is $K(1 - (1 - 1/K)^{chN}) \approx K(1 - e^{-chN/K})$. By union bound, $\mathbb{P}[\exists k : N_k = 0] \leq Ke^{-chN/K}$. $\square$

This theorem establishes that CSVQ's channel aggregation significantly improves codebook utilization compared to vector quantization methods, where utilization scales as $1 - (1 - 1/K)^N$ without the channel multiplier $d$.

## B.5 PROOF OF PROPOSITION (PER-SITE LOOKUP AND MEMORY)

**Proposition (Per-site lookup and memory).** Per-site nearest-neighbor (NN) complexity: VQ-VAE $O(K \cdot ch)$, RQ-VAE $O(D \cdot K \cdot ch)$, Ours $O(ch \cdot K)$ (scalar abs diff). The complexity of each method is proportional to the dimensionality of the comparison target.

For VQ-VAE complexity analysis, we examine the process of computing Euclidean distances between $ch$-dimensional feature vectors $z_e^{(h,w)} \in \mathbb{R}^{ch}$ and $K$ $ch$-dimensional codebook entries $\{c_k\}_{k=1}^{K}$ at each spatial site $(h, w)$:

$$\text{Distance computation}: \|z_e^{(h,w)} - c_k\|_2^2 = \sum_{i=1}^{ch} (z_e^{(h,w,i)} - c_{k,i})^2 \tag{7}$$

$$\text{Per-channel operations}: ch \text{ subtraction + square operations per codebook entry} \tag{8}$$

$$\text{Number of codebook entries}: K \text{ entries to compare} \tag{9}$$

$$\text{Total operations}: K \times ch \text{ operations per site} \tag{10}$$

$$\text{Therefore}: \text{Complexity} = O(K \cdot ch) \tag{11}$$

RQ-VAE complexity analysis is based on the characteristic of performing full vector quantization at each of $D$ sequential residual quantization stages:

$$\text{Stage 1}: \text{Same as VQ-VAE}: O(K \cdot ch) \text{ operations} \tag{12}$$

$$\text{Stages 2 to } D : \text{Each } O(K \cdot ch) \text{ operations on residual vectors} \tag{13}$$

$$\text{Total stages}: D \text{ sequential quantization steps} \tag{14}$$

$$\text{Overall complexity}: D \times O(K \cdot ch) = O(D \cdot K \cdot ch) \tag{15}$$

For CSVQ complexity analysis, we consider the structural characteristic of performing independent one-dimensional scalar distance comparisons for each channel:

$$\text{Distance for channel } i : |\tilde{z}_{e,h,w}^{(i)} - c_k| \text{ for each } k \in [K] \tag{16}$$

$$\text{Per-channel operations}: K \text{ scalar comparisons per channel} \tag{17}$$

$$\text{Total channels}: ch \text{ independent channels} \tag{18}$$

$$\text{Overall complexity}: ch \times O(K) = O(ch \cdot K) \tag{19}$$

Detailed analysis of memory complexity is as follows:

- **VQ-VAE:** Storage of $K$ $ch$-dimensional vectors $\Rightarrow$ Memory = $K \times ch$ scalars = $O(K \cdot ch)$
- **RQ-VAE:** $D$ levels, each with $K$ $ch$-dimensional vectors $\Rightarrow$ Memory = $D \times K \times ch$ scalars = $O(D \cdot K \cdot ch)$
- **CSVQ:** Storage of only $K$ scalar values $\Rightarrow$ Memory = $K$ scalars = $O(K)$

Detailed analysis of the proposed method's practical performance advantages shows that while the theoretical complexity is $O(ch \cdot K)$, asymptotically identical to VQ-VAE's $O(K \cdot ch)$, it provides the following substantial computational advantages:

1. **Operation simplification:** Each operation is simplified to 1D scalar comparison ($|a - b|$), providing faster execution compared to vector inner products $\sum_{i=1}^{ch} (a_i - b_i)^2$
2. **Memory access patterns:** Sequential memory access improves cache efficiency (increased cache hit ratio)
3. **Parallelization capability:** Independent channel processing enables SIMD optimization and GPU parallelization
4. **Numerical stability:** 1D distance computation reduces floating point error accumulation and overflow/underflow risks
5. **Branch prediction:** Simple comparison operations increase CPU branch predictor efficiency

Detailed analysis of CSVQ's space complexity advantages is broken down as follows:

$$\text{Codebook storage:} \quad O(K) \text{ vs. } O(K \cdot ch) \text{ for VQ-VAE} \tag{20}$$

$$\text{Intermediate computation:} \quad O(ch) \text{ vs. } O(K \cdot ch) \text{ for distance matrix} \tag{21}$$

$$\text{Total reduction factor:} \quad \frac{1}{ch} \text{ for codebook, } \frac{1}{K} \text{ for computation} \tag{22}$$

## C  ADDITIONAL ANALYSIS AND EXPERIMENTAL RESULTS

This subsection presents additional theoretical analysis and experimental results that provide deeper insights into CSVQ's behavior.

### C.1  LEARNING STABILITY AND INDEPENDENCE ANALYSIS

**Lemma 3** (Gradient variance reduction). *Let $g_k$ be the gradient estimate for codeword $c_k$ under CSVQ:*

$$g_k = \frac{1}{M_k} \sum_{i=1}^{ch} \sum_{h=1}^{H} \sum_{w=1}^{W} \mathbf{1}[z_{q,h,w}^{(i)} = k] \cdot \psi_{h,w}^{(i)},$$

*where $M_k = \sum_{i,h,w} \mathbf{1}[z_{q,h,w}^{(i)} = k]$ and $\psi_{h,w}^{(i)}$ is the local gradient signal. Under assumptions of 1) weak inter-channel correlation: $|Cov(\psi_{h,w}^{(i)}, \psi_{h',w'}^{(j)})| \leq \delta\sigma^2$ for $i \neq j$ with $\delta \ll 1$, 2) uniform variance: $Var(\psi_{h,w}^{(i)}) = \sigma^2$, and 3) selection independence, we have:*

$$Var(g_k) = \frac{\sigma^2}{M_k} + O(\delta) \approx \frac{K\sigma^2}{chHWB} = \Theta(1/ch)$$

*Proof.* Under weak correlation and independence assumptions, the variance decomposes as $Var(g_k) = \frac{\sigma^2}{M_k} + O(\delta)$. With uniform selection probability $1/K$, we have $\mathbb{E}[M_k] = \frac{ch \cdot H \cdot W \cdot B}{K}$, yielding the $\Theta(1/ch)$ scaling. $\square$

**Theorem 4** (Convergence acceleration). *Under the gradient variance bound from Lemma 3 and standard SGD assumptions with learning rate $\eta$ and $T$ iterations, the optimization error bound for CSVQ scales as:*

$$\mathbb{E}[\mathcal{L}(\theta^{(T)}) - \mathcal{L}(\theta^*)] = O\left(\frac{\sigma^2}{ch\eta T}\right)$$

*where $\theta^*$ is the optimal parameter and $\theta^{(T)}$ is the parameter after $T$ iterations.*

*Proof.* Follows from standard SGD convergence analysis with the improved gradient variance bound $\Theta(1/ch)$ from channel aggregation. $\square$

Full proof is provided in Appendix B.3.

### C.2  TRAINING STABILITY EXPERIMENTS

We evaluate stability using loss and gradient variance across 5 seeds. Protocol P1 uses a 64-parameter budget with VQ-VAE (codebook 16, channels 4) and CSVQ (codebook 64, channels 4), while P4 uses a fixed 256 codebook with 32 channels. These choices follow Section 4.2: in P1 we fix the codebook-parameter budget $M_{cb} := K \times$(codebook dimension) to control memory/search cost for fair comparison (VQ/RQ: 16×4, CSVQ: 64×1 so $M_{cb}$=64), whereas in P4 we fix $(K, d)$ to reflect practical matched configurations used in prior work and to probe stability at a common tokenization capacity; together, they cover tight-resource (P1) and ample-capacity (P4) regimes. Figures 2 and 3 demonstrate CSVQ's superior training stability versus VQ-VAE across both protocols. VQ-VAE exhibits high variance with increasing trends, particularly under P1, with gradient variance showing divergent tendencies and large confidence intervals. Periods where VQ-VAE's confidence intervals collapse to zero indicate codebook collapse (unused entries with vanishing gradients), while CSVQ maintains consistent variance throughout training due to its shared codebook design and channel-wise independence.

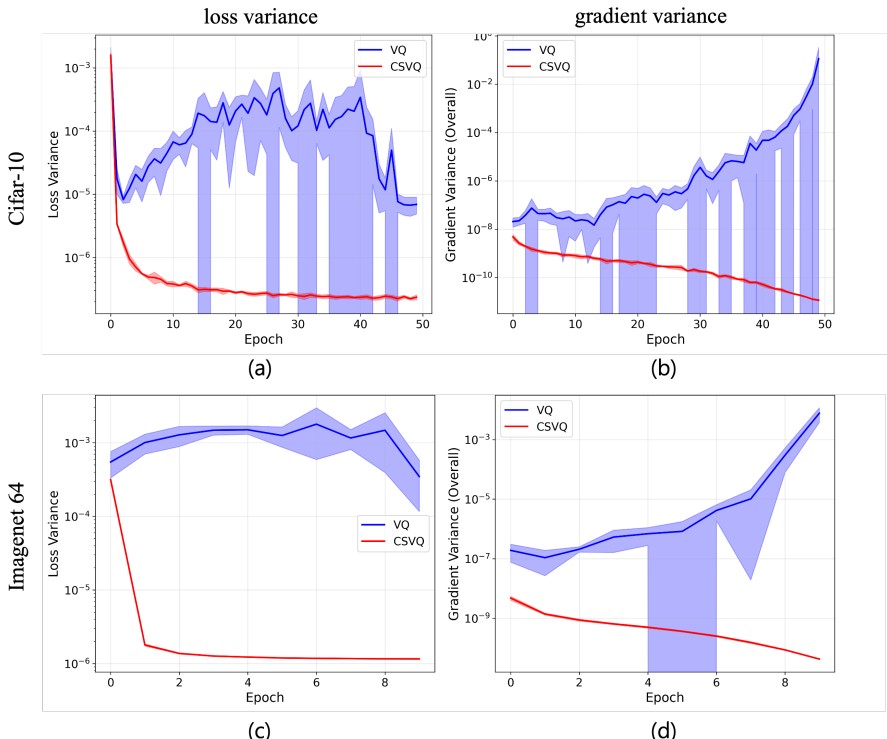

Figure 2: Training stability under P1. Datasets: CIFAR-10 (a,b) and ImageNet-64 (c,d). Panels: loss (a,c) and gradient (b,d) variances. Means are shown as lines; shaded regions denote standard deviations.

### C.3    CODEBOOK UTILIZATION EXPERIMENTS

Protocol P4 analyzes utilization across codebook sizes (8,16,32,64) with varying channels (CIFAR-10: 16, ImageNet-64: 32). We measure perplexity, dead entry ratios, and PSNR performance (see Appendix B.4 for theoretical analysis). Figures 4 and 5 show that CSVQ achieves higher perplexity, lower dead ratios, and superior PSNR performance, indicating superior capacity utilization. CSVQ demonstrates consistent and stable performance with perplexity approaching the codebook size and maintaining steady values throughout training. In contrast, VQ-VAE shows declining perplexity with codebook collapse, particularly severe on ImageNet-64. Dead entry ratios for CSVQ converge to zero across all settings, while VQ-VAE exhibits highly unstable patterns with ratios reaching up to 50% in some configurations.

Regarding PSNR performance, CSVQ shows consistent improvement throughout training and benefits from larger codebook sizes, demonstrating stable learning dynamics. However, the performance improvement diminishes as the codebook size increases, allowing us to empirically identify the optimal codebook size suitable for CIFAR-10. In contrast, VQ-VAE exhibits performance improvements with larger codebooks but suffers from unstable performance during the training process, leading to inconsistent reconstruction quality.

### C.4    ABLATION STUDIES

We analyze the contribution of CSVQ components by ablating (i) the shared codebook (replacing it with per-channel codebooks; "w/o shared codebook"), and (ii) the channel-wise normalization under the shared codebook setting ("w/o normalization"; shared codebook, no normalization). As summarized in Table 7, removing either component can yield slightly higher pixel-space reconstruction (PSNR/SSIM) on CIFAR-10 and ImageNet subset-10, consistent with a classic regularization trade-off. However, the full CSVQ consistently delivers markedly better representation quality in

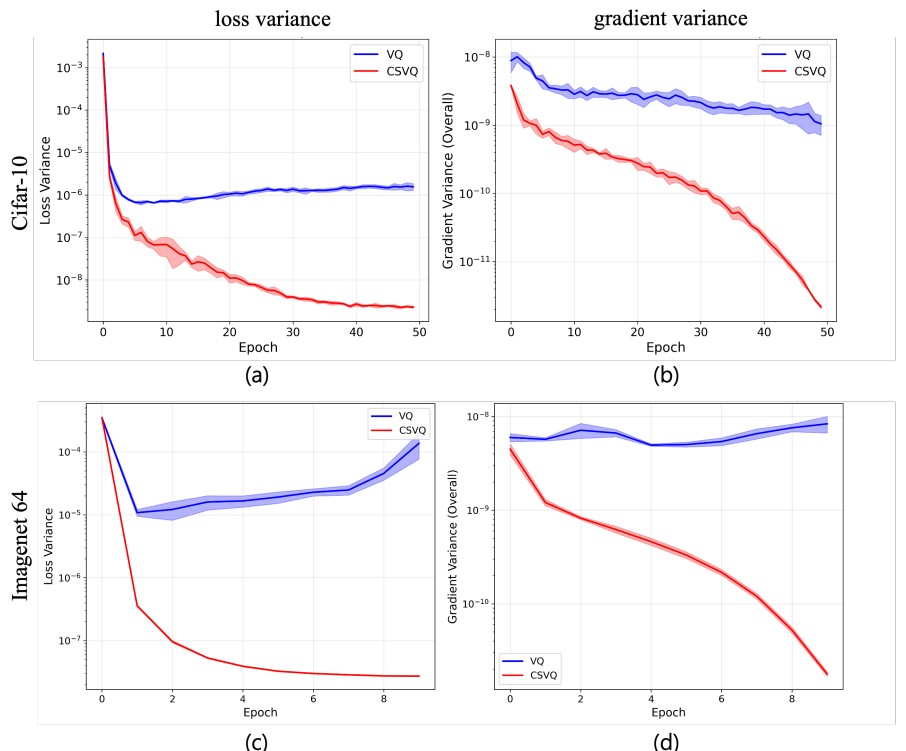

Figure 3: Learning stability under P4. Datasets: CIFAR-10 (a,b) and ImageNet-64 (c,d). Panels: loss (a,c) and gradient (b,d) variances. Means are lines; shaded areas indicate standard deviations.

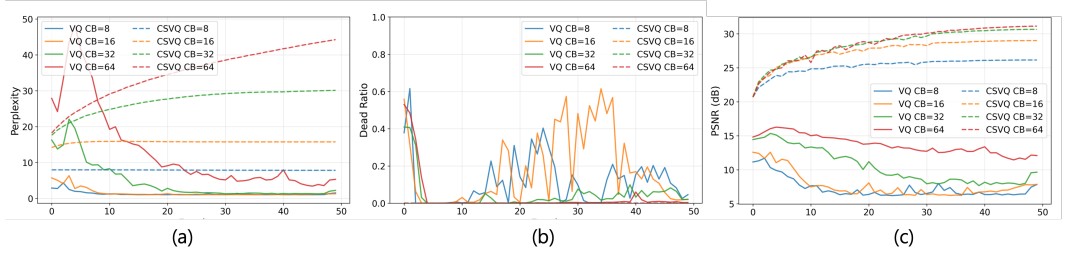

Figure 4: Codebook utilization under P4 (CIFAR-10, $ch = 16$). Panels: (a) perplexity, (b) dead-entry ratio, and (c) PSNR versus $K \in \{8, 16, 32, 64\}$.

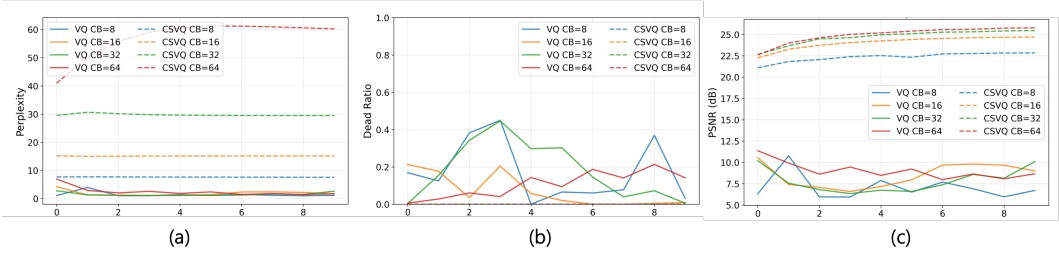

Figure 5: Codebook utilization under P4 (ImageNet-64, $ch = 32$). Panels: (a) perplexity, (b) dead-entry ratio, and (c) PSNR versus $K \in \{8, 16, 32, 64\}$.

linear probing (Top-1), indicating that the shared codebook and channel-wise normalization jointly promote stable, informative discrete representations that generalize better downstream.

Table 7: Ablation study on Protocol P1. Component contributions with codebook size 16 (CIFAR-10) / 32 (ImageNet subset-10), channels 8. Reconstruction and linear probing (Top-1) are jointly reported.

| Dataset | Method | Codebook | Channels | PSNR↑ | SSIM↑ | LP Top-1↑ |
|---------|--------|----------|----------|-------|-------|-----------|
| CIFAR-10 | w/o shared codebook | 16×8 | 8 | **24.49** | **0.9899** | 40.6 |
| | w/o normalization | 16 | 8 | 24.44 | 0.9897 | 39.96 |
| | CSVQ (full) | 16 | 8 | 23.25 | 0.9870 | **46.3** |
| ImageNet subset-10 | w/o shared codebook | 32×8 | 8 | **17.90** | **0.9631** | 55.23 |
| | w/o normalization | 32 | 8 | 17.68 | 0.9600 | 57.96 |
| | CSVQ (full) | 32 | 8 | 17.41 | 0.9525 | **58.32** |

## THE USE OF LARGE LANGUAGE MODELS

**Tool & Version**: Gemini (Google, 2025-09)
**Research Stage**: Generated visualization scripts.
**Writing Stage**: Language editing of author-drafted text for clarity and conciseness.
**Human Oversight**: All outputs reviewed/edited by the authors; authors accept full responsibility for the content.