# OpenReview forum: "CSVQ: Channel-wise Shared-Codebook Vector Quantization for Stable and Expressive Discrete Representations"
_ICLR.cc/2026/Conference — ICLR 2026 Conference Withdrawn Submission_

### Official Review · Reviewer_G3gi · 2025-10-31

**Soundness:** 2
**Presentation:** 2
**Contribution:** 1
**Rating:** 2
**Confidence:** 3

**Summary:**

The paper proposes Channel-wise Shared-codebook Vector Quantization (CSVQ), a tokenization method that performs independent scalar quantization on each latent channel using a single shared codebook, preceded by channel-wise normalization. By quantizing channels independently while sharing a single codebook across all channels, CSVQ achieves the same representational capacity as vector-quantized latents but with a much smaller codebook size.
Experiments on CIFAR-10 and ImageNet-64 show that CSVQ improves reconstruction quality across three experimental settings: fixed codebook budget, equal expressiveness, and channel scaling.

**Strengths:**

The presentation is clear and straightforward, making the paper easy to follow.

**Weaknesses:**

**Limited Novelty**:

- Channel-wise quantization with normalization: The core idea is similar to MoVQ [1] (Section 3.1, Multichannel Representation). Classic VQ partitions the latent space only along the spatial grid, producing discrete codes in $\mathbb{R}^{H×W}$. The proposed method additionally splits the channel dimension into sub-spaces, yielding codes in $\mathbb{R}^{H×W×ch}$.

  The key difference is that CSVQ quantizes each channel value independently (i.e., sub-space dimension = 1), whereas MoVQ uses higher-dimensional sub-spaces. While this approach naturally increases latent capacity and improves reconstruction fidelity, it also significantly increases the code sequence length from $H×W$ to $H×W×ch$. This longer sequence makes autoregressive generation (for image generation from discrete tokens) computationally expensive and potentially more difficult to model effectively. This may explain why the authors did not conduct image generation experiments, which are a standard benchmark for VQ-based methods.

- The theoretical contribution presented in the main text appears straightforward. Using more codes to represent an image while maintaining the same codebook size increases representational capacity is intuitive and directly leads to the theoretical results presented. The analysis does not provide substantial new theoretical understanding beyond this obvious observation.

**Limited Experimental Scope**:

- Experiments are restricted to CIFAR-10 and ImageNet-64, which are relatively small-scale datasets.
- Missing image generation benchmark: The paper does not include image generation experiments, which are a common and critical application of VQ-based methods.

**Questions:**

See the weaknesses above.

---

### Official Review · Reviewer_HQdg · 2025-10-31

**Soundness:** 2
**Presentation:** 1
**Contribution:** 1
**Rating:** 2
**Confidence:** 4

**Summary:**

The paper proposes Channel-wise Shared-codebook Vector Quantization (CSVQ) which use one shared scalar codebook to quantize each channel independently. The authors claim CSVQ delivers the expressiveness of vector VQ with a much smaller codebook, reduces gradient variance via cross-channel signal aggregation, and scales well in channels. To demonstrate it, the paper provides experiments on CIFAR-10 and ImageNet-64 and reports superior PSNR/SSIM to show better reconstruction quality. Additionally, the research offers better linear-probe and transfer results than VQ-VAE/RQ-VAE/FSQ/LFQ under the evaluated protocols.

**Strengths:**

1. This paper explores an alternative direction to improve the vector quantization technique.
2. The authors provide both theoretical and empirical analyses.

**Weaknesses:**

1. Novelty
    1. There are some concerns about the novelty of this paper, as [1] has already proposed a similar channel-wise approach.
2. Experiment:
    1. The experiments are not conducted under standard settings [2, 3, 4, 5, 6, 7, 8], which raises doubts about CSVQ’s effectiveness in real-world scenarios and about the fairness of the comparisons among baselines.
        1. Dataset. For image tokenizer, a standard setup typically uses ImageNet-1k with the resolution of 256 * 256 or 512 * 512.
        2. Codebook & Dimension settings.
            1. The latent bottlenecks are overly tight (like codebook for 64 and dimension for 1 or 4). That is not reasonable or applicable in a real-world scenario. A more applicable configuration would scale the latent space(like codebook size 512 with 32/64 dimensions on CIFAR-10[9]).
            2. For the experiment, which set the baseline’s channel dimension as 1, [8] would be a useful and meaningful baseline for comparison.
        3. Metrics. It is better to report codebook usage, Perplexity, LPIPS, and rFID score, which are commonly used in other baselines for a fair comparison.
        4. Optimization targets. For image tokenizers, a standard setting is often based on VQGAN[6], where the final objective includes quantization loss and MSE, as well as GAN loss and perceptual loss (e.g., LPIPS) to emphasize perceptual quality and optimize the latents differently.
        5. Downstream task. Image tokenizers are usually trained for generative models, yet there are no generation experiments on real-world datasets.
3. Lack of support for claims:
    1. The paper mentioned CSVQ can “reduce the gradient noise through channel-wise aggregation”, but there is no theoretical analysis or empirical analysis about it.
    2. The paper mentioned CSVQ is “a simple, stable channel-wise scalar quantizer with a shared codebook that alleviates collapse”, but there is no evidence about the mitigation of collapse.
    3. The CSVQ is “a simple, stable channel-wise scalar quantizer”, but there is also a lack of theoretical comparison about other scalar quantizers like LFQ and FSQ.
4. Concerns about “Expressiveness”:
    1. There is also a concern about the theoretical analysis, especially about the expressiveness. For VQ and RVQ, the usual setting for ch is not 1. That means the real expressiveness for VQ and RVQ should be larger than the cardinality of the symbol space |S|. Thus, the theoretical analysis framework has a preference for CSVQ and unfariness for VQ or RVQ.
    2. The current theoretical expressiveness analysis hinges on independent per-channel selection, but real latents might not be independent across channels. The configuration counting is thus an upper bound, not realized capacity.
5. Writing:
    1. There is no appendix in the paper, despite being referenced multiple times.
    2. For Equations (210–211), (215–216), (227–232), etc., the equation numbers are missing.
    3. At the ends of lines 260, 269, and 280, there are odd square markers.

[1] Yang, Tong, Xiangyu Zhang, and Wenqiang Zhang. "Image Generation with Channel-wise Quantization.”

[2]Zhu, Yongxin, et al. "Addressing representation collapse in vector quantized models with one linear layer." *Proceedings of the IEEE/CVF International Conference on Computer Vision*. 2025.

[3]Mentzer, Fabian, et al. "Finite scalar quantization: Vq-vae made simple." *arXiv preprint arXiv:2309.15505* (2023).
[4]Sun, Peize, et al. "Autoregressive model beats diffusion: Llama for scalable image generation." *arXiv preprint arXiv:2406.06525* (2024).
[5]Yu, Jiahui, et al. "Vector-quantized image modeling with improved vqgan." *arXiv preprint arXiv:2110.04627* (2021).
[6]Esser, Patrick, Robin Rombach, and Bjorn Ommer. "Taming transformers for high-resolution image synthesis." *Proceedings of the IEEE/CVF conference on computer vision and pattern recognition*. 2021.

[7]Kouzelis, Theodoros, et al. "Eq-vae: Equivariance regularized latent space for improved generative image modeling." *arXiv preprint arXiv:2502.09509* (2025).

[8]Bachmann, Roman, et al. "FlexTok: Resampling Images into 1D Token Sequences of Flexible Length." *Forty-second International Conference on Machine Learning*. 2025.

[9] Van Den Oord, Aaron, and Oriol Vinyals. "Neural discrete representation learning." *Advances in neural information processing systems* 30 (2017).

**Questions:**

1.  It would be beneficial if authors could  include an ablation study that increases the latent size and shows the corresponding reconstruction improvements, to validate that CSVQ is a general method across different latent configurations.
2. It would be great if authors could consider addressing the points raised in the weakness section

---

### Official Review · Reviewer_EbtN · 2025-11-01

**Soundness:** 3
**Presentation:** 2
**Contribution:** 2
**Rating:** 2
**Confidence:** 5

**Summary:**

This paper presents Channel-wise Shared-codebook Vector Quantization (CSVQ), a new tokenizer that addresses the stability and efficiency issues of traditional VQ methods. Instead of quantizing entire vectors with large, complex codebooks, CSVQ quantizes each latent channel independently using a single, shared scalar codebook. The approach achieves exponential representational capacity with memory cost $O(K)$. Empirical results on CIFAR-10 and ImageNet-64 show that CSVQ outperforms existing VQ methods in reconstruction quality, stability, and downstream task performance.

**Strengths:**

- The proposed method, CSVQ demonstrates excellent performance across reconstruction quality (as shown in Tables 2, 3, and 4), but also excels in downstream tasks such as linear probing and transfer learning (Table 6).
- By sharing a single scalar codebook across all channels, CSVQ improved codebook utilization. This makes the model more reliable, easier to train, and more efficient in its use of learned parameters.

**Weaknesses:**

- The paper's central claim of achieving high representational capacity ($K^{ch \times H \times W}$) by factoring in the channel dimension ($ch$) is questionable. The total number of bits required to represent an image is what truly matters. This is typically calculated by `(number of tokens) × log₂(codebook_size)`. For instance, in Table 3, CSVQ uses $8 \times 8 \times 4$ tokens, each from a codebook of size 64. A fair comparison would require baseline methods (like FSQ, LFQ) to use a configuration that yields a similar total bitrate. For example, a single spatial grid of $8 \times 8$ tokens with a codebook of size $64 ^ 4$, or other equivalent configurations.

- The proposed tokenizer creates an impractically large discrete space for generative modeling. For example, in Table 4, with $K=512$ and $ch=3$, the vocabulary size is $512^3 \approx 134$ million. Training an autoregressive model (like a Transformer) on such a massive vocabulary is computationally infeasible with current methods. This severely limits the practical impact of CSVQ as a general-purpose tokenizer for generative tasks, which is a primary application for VQ methods.

- The experimental section, while extensive, lacks sufficient qualitative and analytical depth. For instance, Table 6 shows that CSVQ leads to better downstream performance, but the paper offers no investigation into *why*. What specific features has the encoder learned that make the representations more semantically meaningful? Visualizations of the learned features, analysis of the per-channel statistics after normalization, or feature attribution maps would have provided much-needed insight into the model's behavior and could have strengthened the claims about its superior understanding capabilities.

- The paper is missing critical ablation studies that would justify its specific design choices. A key component of CSVQ is the per-channel normalization step applied before quantization. However, its importance is not experimentally validated. An ablation study comparing CSVQ with and without this normalization step is essential to demonstrate its contribution to performance and stability. The paper discusses the trade-offs between codebook size ($K$) and channel depth ($ch$), but this relationship is not visually or comprehensively analyzed. A graphical analysis showing how reconstruction quality changes as $K$ and $ch$ are varied would provide a much clearer picture of these trade-offs.

**Questions:**

- The paper positions CSVQ as a "tokenizer" but does not evaluate its performance in a generative setting (e.g., training a Transformer on the quantized latents to generate new images). How does CSVQ perform on standard image generation benchmarks compared to other VQ methods?

---

### Official Review · Reviewer_L2Bx · 2025-11-01

**Soundness:** 2
**Presentation:** 3
**Contribution:** 2
**Rating:** 4
**Confidence:** 3

**Summary:**

This paper addresses the issues of training instability and large codebook memory in Vector Quantization (VQ) models. This paper introduces CSVQ, which quantizes each latent channel independently using a single, shared scalar codebook with per-channel normalization. Extensive experiments and theoretical analysis demonstrate that CSVQ achieves comparable representation capacity with a smaller codebook.

**Strengths:**

1.Reducing gradient variance by roughly 1/ch where ch is the number of latent channels, thus enhancing training stability.

2. The experiments on reconstruction tasks are convincing and the results on linear probing are interesting, suggesting that CSVQ acquires a degree of semantic understanding.

3.Maintaining comparable performance with smaller codebook size.

**Weaknesses:**

1.Missing Generation Results: A core objective of VQ-VAE models is to learn meaningful discrete representations that are applicable to generation tasks. Therefore, it is meaningful and necessary to present generation results to thoroughly evaluate the quality and effectiveness of CSVQ. It is currently unclear whether CSVQ can be effectively adapted to generative models such as autoregressive models or diffusion large language models (dllms). In contrast, patchify-based VQ codebooks, leveraging the spatially local relationships between image patches, have been widely validated and demonstrated effectiveness in both autoregressive models and dllms.

2.Missing Baselines: There exits other methods that aim to solve training instability, including eqvae[1] and rotation trick[2].

3.Limited results and metrics:Despite emphasizing smaller codebook sizes, experiments on the widely-used ImageNet 256 dataset and the comparisions between large cobebook size are absent. Additionally, the paper claims better codebook usuage, however codebook usage rate (as in SimVQ) is not reported.

[1] Kouzelis T, Kakogeorgiou I, Gidaris S, et al. Eq-vae: Equivariance regularized latent space for improved generative image modeling[J]. arXiv preprint arXiv:2502.09509, 2025.
[2] Fifty, Christopher, et al. "Restructuring Vector Quantization with the Rotation Trick." ICLR. 2025.

**Questions:**

1.A significant concern is the scalability of CSVQ for high-resolution image where the dimension length in the codebook is equal to h*w. For example, an 8x downsampling encoder of a 512x152 image yields a 4096-dimensional representations for each channel in the CSVQ setting. However, in traditional VQ-VAE, the length of the dimension is often less than 128 or even 8 [3] . It is crucial to clarify if CSVQ can achieve superior performance with a comparatively smaller codebook size (K) while handling this long demension length, and how its efficiency is impacted.

2.Please clarify why sharing the same codebook across all channels, can improve stability in line 57-58.

[3] Sun P, Jiang Y, Chen S, et al. Autoregressive model beats diffusion: Llama for scalable image generation[J]. arXiv preprint arXiv:2406.06525, 2024.

---

### Note · Authors · 2025-11-22

I have read and agree with the venue's withdrawal policy on behalf of myself and my co-authors.